# Evaluation of miniaturized Illumina DNA preparation protocols for SARS-CoV-2 whole genome sequencing

Sureshnee Pillay[1]*, James Emmanuel San[1,2], Derek Tshiabuila[1], Yeshnee Naidoo[1,2], Yusasha Pillay[2], Akhil Maharaj[1], Ugochukwu J. Anyaneji[1,2], Eduan Wilkinson[2], Houriiyah Tegally[1,2], Richard J. Lessells[1,3], Cheryl Baxter[2,3], Tulio de Oliveira[1,2,3,4]*, Jennifer Giandhari[1]*

**1** KwaZulu-Natal Research Innovation and Sequencing Platform (KRISP), Nelson R Mandela School of Medicine, University of KwaZulu-Natal, Durban, South Africa, **2** Centre for Epidemic Response and Innovation (CERI), School of Data Science and Computational Thinking, Stellenbosch University, Stellenbosch, South Africa, **3** Center for AIDS Programme of Research in South Africa (CAPRISA), Durban, South Africa, **4** Department of Global Health, University of Washington, Seattle, WA, United States of America

* pillaysureshnee@gmail.com (SP); Giandharij@ukzn.ac.za (JG); tulio@sun.ac.za (TO)

**Data Availability Statement:** All the raw sequence data generated for this research are available publicly in the NCBI Sequence Read Archive (project no. PRJNA926488). The methods follow to

## Abstract

The global pandemic caused by SARS-CoV-2 has increased the demand for scalable sequencing and diagnostic methods, especially for genomic surveillance. Although next-generation sequencing has enabled large-scale genomic surveillance, the ability to sequence SARS-CoV-2 in some settings has been limited by the cost of sequencing kits and the time-consuming preparations of sequencing libraries. We compared the sequencing outcomes, cost and turn-around times obtained using the standard Illumina DNA Prep kit protocol to three modified protocols with fewer clean-up steps and different reagent volumes (full volume, half volume, one-tenth volume). We processed a single run of 47 samples under each protocol and compared the yield and mean sequence coverage. The sequencing success rate and quality for the four different reactions were as follows: the full reaction was 98.2%, the one-tenth reaction was 98.0%, the full rapid reaction was 97.5% and the half-reaction, was 97.1%. As a result, uniformity of sequence quality indicated that libraries were not affected by the change in protocol. The cost of sequencing was reduced approximately seven-fold and the time taken to prepare the library was reduced from 6.5 hours to 3 hours. The sequencing results obtained using the miniaturised volumes showed comparability to the results obtained using full volumes as described by the manufacturer. The adaptation of the protocol represents a lower-cost, streamlined approach for SARS-CoV-2 sequencing, which can be used to produce genomic data quickly and more affordably, especially in resource-constrained settings.

## Introduction

Since the emergence of coronavirus disease-2019 (COVID-19), whole genome sequencing (WGS) of SARS-CoV-2 has become a critical tool for tracking viral evolution and

generate the sequences have been published on protocols.io (dx.doi.org/10.17504/protocols.io.n92ldpp8nl5b/v1). All primers used to generate the sequences have been provided in S2 Table.

**Funding:** Rockefeller Foundation, HTH 017, Prof Tulio de Oliveira National Institutes of Health, U01 AI151698 for the United World Antivirus Research Network (UWARN) and the INFORM Africa project through IHVN, U54 TW01204 - Prof Tulio de Oliveira African Society for Laboratory Medicine, Prof Tulio deOliveira H3BioNetAfrica, 2020 HTH 062, Prof Tulio de Oliveira Department of Science and Innovation, South Africa, ProfTulio de Oliveira and the South African Medical Research Council, Prof Tulio deOliveira Abbott Pandemic Defense Coalition (APDC) - Prof Tulio de OliveiraWorld Bank (TF0B8412) - Prof Tulio de Oliveira South African Medical Research Council, SAMRC SIR-HIVDR POC, Dr JenniferGiandhari

**Competing interests:** The authors have declared that no competing interests exist.

understanding the virus's evolutionary dynamics during disease outbreaks [1–5]. WGS has been widely used to provide additional data to complement routine diagnostic testing by enabling the surveillance of SARS-CoV-2 variants as well as the identification of new variants. The information gained from analyses of viral WGS serves to inform public health responses by describing transmission dynamics of disease outbreaks. Throughout the COVID-19 pandemic, next-generation sequencing has been primarily used to produce whole genomes to characterize the virus, to understand its contagiousness and pathogenic features, and most of all to track its evolution [6]. As of November 2022, > 11 million SARS-CoV-2 sequences have been produced and shared globally [7].

Although several sequencing technologies exist today, Illumina and Nanopore ONT represent the two most used technologies throughout the pandemic. Both technologies offer significant advantages that underpin their relative importance. Nanopore is able to produce long reads that encompass all of the mutations within a single virus particle. The technology is also relatively cost-effective, portable and provides sequence data in real-time. The major drawback of Nanopore sequencing is the higher error rates as compared to Illumina sequencing [8]. Illumina produces short reads albeit with a low error rate as well as high throughput sequencing [9]. As such, although Nanopore is cheaper and faster, Illumina remains the most dominant sequencing technology for the production of high-quality whole genome sequences needed for genomic surveillance.

Typically, The number of samples that can be sequenced is determined by both the cost and availability of reagents. However, the disruption caused by the pandemic to reagent manufacturing, the logistics of reagent delivery as well as the global demand for sequencing resulted in temporary shortages and delivery delays in many countries. As a result of limited supplies, protocol modification has been adopted as means to maximize reagent use [10] and drive down sequencing costs. Despite all the efforts towards reduction in sequencing costs, the cost of sequencing has been decreasing [11], and the cost of library preparation remains a major bottleneck [12]. Instrument-specific reagent kits make up approximately 75% of the sequencing cost. This underpins the importance of the library preparation step towards sequencing cost reduction. The most efficient way to reduce the cost of library preparation is to reduce the reaction volume [12]. The miniaturisation of reagent volumes as well as maximising the number of samples that can be sequenced per run can significantly reduce the cost of sequencing per sample. The reagent kits can only be applied to a single run and cannot be modified. Maximising the number of samples can be achieved by using a larger number of unique indexes so more samples can be multiplexed. The miniaturisation of reagent volumes has been successfully explored previously for various methods. Minich et al were able to reduce their cost of 16s RNA by 58% without compromising sequencing success [13]. Orgiso-Tanaka showed that their method for Ampliseq library preparation saved 86.8% in reagent usage, confirming that miniaturization could reduce the cost of AmpliSeq library preparation [12]. For RNAseq Mayday et al showed in 384-well mNGS library preparations, they achieved cost savings of over 80% in materials and reagents alone, and reduced preparation time by 90% compared to manual approaches, without compromising quality or representation within the library [14]. While Fuchs et al showed that the Mini-XT protocol maintains sequence quality while reducing library preparation reagent volumes eightfold for SARS-CoV2 [15]. Although these methods primarily utilized automated liquid handling systems, together they highlight the feasibility and potential of miniaturization to lower sequencing costs and maximise reagent use in resource-limited settings.

Modified library preparation methods can be validated with acceptable quality metrics. However, for most researchers, this remains challenging especially if there are financial constraints. Given the cost and stock limitation experienced during the SARS-CoV-2 pandemic,

in this study, an adaption of our library preparation protocol into a lower-cost, streamlined protocol is described. The results show that even without the high cost of automated liquid handlers commonly used in library preparation, the method could still be implemented manually. Additionally, a detailed description of the cost and time efficiency of the method together with its quality and performance metrics using the Illumina DNA Prep Library Preparation Kit are provided.

## Materials and methods

### Sample selection and ethics approval

Forty-seven remnant nasopharyngeal swab samples were obtained from the National Health Laboratory Service as part of a SARS-CoV-2 genomic surveillance programme. Samples were selected for sequencing based on a positive qPCR result, irrespective of the cycle-threshold (Ct) value of the result. This study falls under the approval of the Biomedical Research Ethics Committee of the University of KwaZulu-Natal, South Africa, with protocol reference number BREC/00001510/2020. As no human subjects were involved, no informed consent was required.

### Sample preparation

RNA was extracted from the samples on the automated Chemagic 360 instrument, using the CMG-1049 kit (Perkin Elmer, Hamburg, Germany) and was stored at -80˚C until further use.

### cDNA and tiling-based polymerase chain reaction

Complementary DNA (cDNA) synthesis was performed using LunaScript—RT SuperMix kit (New England Biolabs), where 1 μl of LunaScript and 5.5 μl of RNA were used, followed by gene-specific multiplex PCR, using the ARTIC protocol V1 [16], as described previously [17]. The volumes of reagents used are summarised in Table 1. A master mix was made for each of the two pools which contained a multiplex of SARS-CoV-2 primers (S2 Table). The ARTIC protocol uses two pools of primers which are continuously being improved to amplify ~400bp tiled amplicons [15]. SARS-CoV-2 whole-genome amplification by multiplex PCR was attempted using V4 primers designed on Primal Scheme (http://primal.zibraproject.org/) to generate 400 base pair (bp) amplicons with 70 bp overlaps, covering the 30 kilobase SARS-CoV-2 genome. Random PCR products were quantified using the Qubit double-strand DNA (dsDNA) High Sensitivity assay kit on a Qubit 4.0 instrument (Life Technologies).

### Illumina DNA Prep library

The Illumina DNA Prep Kit (Illumina, San Diego) was tested using four different methods. The first was according to the manufacturer's instructions, the second was a full reaction using

**Table 1. Reagent volumes used for tiling PCR.**

| Component | Pool 1 and 2 volumes (μl) |
|---|---|
| 5X Q5 Reaction Buffer | 2.5 |
| 10mM dNTPs | 0.25 |
| Q5 Hot Start DNA Polymerase | 0.125 |
| Primer Pool 1 or 2 (10μM) | 1.8 |
| Nuclease free water | 5.4 |
| *Total* | *10* |

**Table 2. Comparison of the manufacturer-recommended reagent volumes for major steps and reagent volumes used for the miniaturised protocols.**

|  | Manufacturer recommended (Full reaction) | Full reaction rapid | Half reaction | One-tenth reaction |
|---|---|---|---|---|
| DNA Amplicon (ul) | 10 | 10 | 5 | 5 |
| Tagmentation mix (ul) | 20 | 20 | 10 | 2 |
| EPM (ul) | 40 | 40 | 20 | 4 |
| Indexes (ul) | 10 | 10 | 5 | 1 |

the rapid method (described below), the third was a half-reaction using the rapid method, and the fourth was a one-tenth reaction using the rapid method as described in protocols.io [18].

For each approach 47 of the same samples were tested. The reagent volumes used for each parameter are provided in Table 2. To manage the cost of sequencing, we split for methods across two sequencing runs. Each run accounted for two methods each having 47 samples and a control. Importantly, the method of interest in this manuscript has since been used across several sequencing runs successfully which gives us further confidence in its performance and accuracy.

The four different libraries were run on two different Miseq instruments, two libraries were run together on one instrument and the other two libraries were run together on another instrument.

## Full reaction full method as per manufacturer's instructions

Briefly, undiluted tiling PCR amplicons were used. The DNA was tagmented with bead-linked transposomes, and the tagmentation reaction was stopped with a tagmentation stop buffer before proceeding to the post-tagmentation clean-up, using the tagmentation wash buffer. This step was followed by eight cycles of amplification of the tagmented DNA with enhanced PCR mix and index adapters. The Nextera DNA CD (Illumina) indexes were used. The libraries were cleaned using sample purification beads, followed by two ethanol washes and eluted in 13.0 ul of resuspension buffer. Libraries were quantified by using the Qubit dsDNA High Sensitivity assay kit on a Qubit 3.0 instrument (Life Technologies). Each sample library was normalized to 4 nM concentrations before being pooled and denatured with 5 μl of 0.2 N sodium acetate. The 12 pM library was spiked with 1% PhiX control (PhiX Control v3) and sequenced on an Illumina MiSeq platform (Illumina), using a MiSeq V2 Reagent Kit (500 cycles).

## Rapid method using full reaction, half-reaction and one-tenth reaction

To reduce the time taken, a rapid clean-up approach was tested which involved pooling the 48 samples before the SPB (Sample Preparation Bead) clean-up (Table 3). Following enrichment, 4.5μl of each library was pooled together giving a total of 216μl. 216μl sample preparation beads and 192 μl nuclease-free water was added to the pooled amplicon library. This was then vortexed and incubated for 5 minutes before being placed on a magnetic stand for approximately 3 minutes. The supernatant was then transferred to a clean tube and 72μl (1.5ul multiplied by 48 samples) of sample preparation beads was added, vortexed and incubated for 5 minutes. This was then placed on the magnetic stand for approximately 3 minutes before washing twice with 1000μl of freshly prepared 80% ethanol. At the end of the wash, all ethanol was removed and eluted in 35μl of resuspension buffer and incubated for 2 minutes before transferring the eluate to a new tube. The pooled amplicon library (PAL) was quantified using the Qubit dsDNA High Sensitivity assay kit on a Qubit 3.0 instrument (Life Technologies). Final libraries were analyzed on the LabChip® GX Touch™ (Perkin Elmer). The PAL was then

**Table 3. Comparison of the rapid and non-rapid methods.**

| Steps | Rapid | Non-Rapid | Comment |
|---|---|---|---|
| First bead Clean | x | x | Pooled clean-up_R vs 48 individual sample clean-up for NR |
| Second bead clean | x | x | Pooled clean-up_R vs 48 individual samples_NR |
| Ethanol wash | x | x | One pool_R vs 48 individual samples_NR |
| Elution | x | x | One pool_R vs 48 individual samples |
| Quantification | x | x | One pool_R vs 48 individual samples |
| Normalization to 4nM | x | x | One pool_R vs 48 individual samples |
| Pooling 4nM libraries |  | x | R is already pooled vs Pooling 48 individual samples_NR |

NR- Non-Rapid

R- Rapid

diluted to 4nM concentration before denaturation and loading on the Miseq as per the full reaction full method above.

The volume of input DNA was kept the same for the half-reaction as well as the one-tenth reaction (5μl) to increase sequencing success for samples with low viral loads. All other reagents were reduced proportionally (Table 2). The full-reaction full method and the half-reaction rapid methods were run together on one Miseq reagent kit while the full-reaction rapid and the one-tenth rapid methods were run together on another Miseq reagent kit. For the half-reaction and one-tenth reaction, the rapid approach was taken with the volumes listed in Table 2.

## Cost and time comparison

To reduce the length of the protocol, a modified version of the Illumina DNA library preparation was tested by the elimination of the quantification of all samples before library preparation and pooling all samples before bead clean-up. This resulted in a single tube of libraries that were purified and quantified on the Qubit instead of 96 samples.

## Sequence assembly and QC

Paired-end Fastq reads were assembled using Genome Detective version 1.132 (https://www.genomedetective.com), which was updated for the accurate assembly and variant calling of tiled primer amplicon Illumina or Oxford Nanopore reads, and the Coronavirus Typing Tool [19, 20]. QC metrics were generated using Nextclade, a tool that identifies differences between your sequences and a reference sequence used by Nextstrain [21], which uses these differences to assign your sequences to clades, and reports potential sequence quality issues in your data. The sequences were then manually edited in Geneious Prime Version 2021.2 to remove or mask low-quality mutations including indels that resulted in frameshifts. The resulting sequences were used to infer a maximum likelihood phylogenetic tree relative to the Wuhan-Hu-1 reference strain, NC_045512.2 (Accession number: MN908947.3) using the GTR evolutionary model as implemented in IQTree2. The tree was inferred using 100 bootstrap replicates to establish branch support.

## Analysis of sequencing metrics

To compare the quality of sequencing between the two methods, consensus QC reports were generated using Nextclade [22]. We compared the genome coverages (as a % of the length of the SARS-CoV-2 reference), the number of missing and mixed bases (Ns), and the number of

private mutations present in the consensus sequences as proxies for sequencing success and quality. Statistical significance was tested using Kruskal-Wallis test to compare the different experimental conditions. The notation Ns indicates non-significant differences. Data visualization was performed in the R programming language using custom scripts [23].

## Results

### Cost and time comparison

NGS library was constructed using the Illumina DNA Prep library kit and CD indexes (Illumina, San Diego). This modification to the protocol presented here, from cDNA synthesis to loading onto the Miseq, significantly reduced the preparation time from 6.5 hrs to 3 hrs, thereby reducing the turnaround time by 3.5 hrs (Fig 1A).

This rapid, miniaturised high-throughput method significantly reduced the library preparation reagent cost associated with sequencing. The reagent cost, as of November 2022, using the full protocol as per the manufacturer's instructions, was 75 USD (Table 4). The full reaction, the rapid method helped reduce time, while using the miniaturized and rapid protocol, the cost per sample dropped to approximately 8.4 USD resulting in almost a 9-fold reduction in costs.

The concentration of the final libraries was quantified using the Qubit 4.0 instrument. For the full reaction, as per the manufacturer's protocol, the samples were quantified individually and normalised to 4nM which resulted in an average of 22.6ng/ul (range: 2.7ng/ul– 32.0ng/ul). For the full reaction rapid method, the final pooled library had a concentration >100ng/ul.

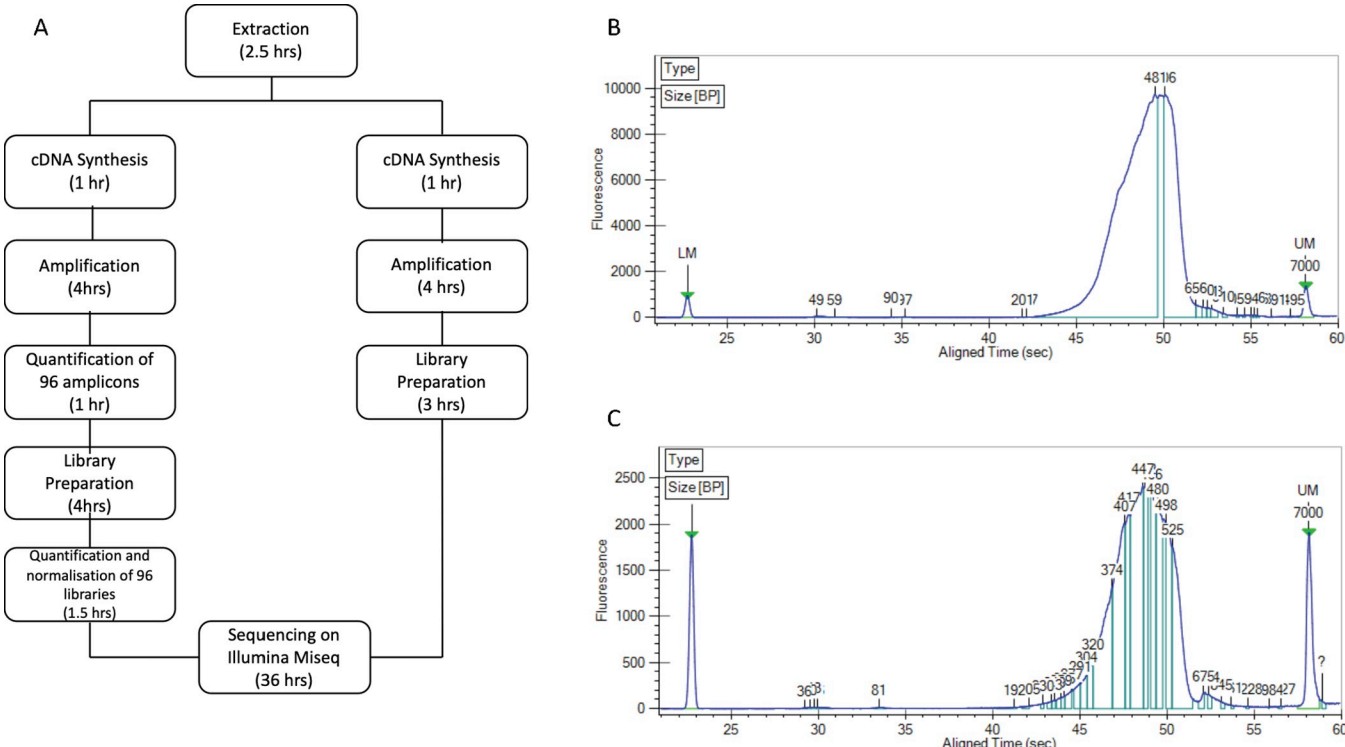

**Fig 1.** **A.** Workflow using the Illumina DNA library preparation kit for both the full method as per manufacturer's instructions (left) as well the adapted rapid method (right). An overall reduction of 3.5 hours was achieved using the adapted rapid method. **B.** Fragment analysis of the 4nM pool of the full reaction, full method. **C.** Fragment analysis of the pooled amplicon library of the one-tenth reaction, rapid method.

**Table 4. Breakdown of cost.**

|  | Rands | USD | Per sample |
|---|---|---|---|
| Full | 103 500 | 7182 | 75 |
| Full rapid | 103 500 | 7182 | 75 |
| Half rapid | 51 750 | 3601 | 37.5 |
| One tenth | 10 350 | 808 | 8.4 |

Similarly, a concentration of >100ng/ul was obtained for the one-tenth reaction method and a slightly lower concentration of 77.6ng/ul was achieved using the half-reaction method (Table 5).

The fragment size of the library of the full reaction full method and the tenth reaction were similar (Fig 1B and 1C). The average fragment size of Illumina DNA Prep libraries is 300-350bp as per the manufacturer's protocol. Even with the miniaturized method, the size of the fragment remained the same.

## Sequencing success and quality comparison

Genomic epidemiology and phylogenetic inference heavily depend on the quality of the consensus sequences. Near-complete genomes enable us to reconstruct evolutionary histories with high confidence [24]. The average genome coverage was relatively high for all four methods with the lowest (97.1%) coming from the half-reaction rapid method (Table 6). The majority of the sequences had a coverage between 95 and a 100% as shown in Fig 2. Illumina recommends a quality score (Q30) >75%. Both sequencing runs were of high quality having a Q30 score of 82% and 78.4%.

Similarly, there was no significant difference in the Ns (unassigned bases) across the four methods (Figs 3 and 4). The number of Ns in the sequencing was also comparable between the full reaction full method and the one-tenth method. Interestingly, all methods suffered N's in the same region of the genome suggesting that it was not a methodological problem but rather lower genome quality in these regions. The sequences generated were categorised as lineages Delta (n = 45 / 47), Beta (n = 1 / 47) and 20D (n = 1 /47). The sequencing results show a high level of congruence between the different reactions. There was only one difference in lineage assignment between the different reactions (See S1 Table, S1 Fig-Tree)

## Discussion

The use of WGS has become an important aspect of the international response to the COVID-19 pandemic [25] and will be the focus of future pandemics [26, 27]. With the ever-increasing demands of WGS, it has become necessary to look at ways of reducing the cost of library preparation methods. Miniaturization techniques have been widely implemented to reduce the cost of sequencing [11, 12, 26].

**Table 5. Qubit concentrations of all four methods.**

| Method | Concentration (ng/μl) |
|---|---|
| Full reaction (full method) | 22.6* (range 2.7–32.0) |
| Full reaction (rapid method) | >100 |
| Half reaction (rapid method) | 77.6 |
| One-tenth Reaction (rapid method) | >100 |

*Mean concentration of 4nM libraries

**Table 6. Sequencing metrics.**

| | Full Reaction | Full Rapid | Half | Tenth |
|---|---|---|---|---|
| Mean number of missing bases (Ns) | 517.3 | 649.4 | 824.5 | 528.2 |
| Mean sequencing Coverage (%) | 98.2 | 97.5 | 97.1 | 98.0 |

Unlike previously described methods [12, 13, 15] which used automated liquid handlers, we embarked on a manual miniaturized method. Previous studies looked at miniaturisation methods for assays such as RNAseq [14], Ampilseq library preparation [12], 16S RNA amplicon preparation [13], and RNAseq library preparation [26]. This has not been previously shown as not many laboratories are comfortable with pipetting such low volumes which could lead to discrepant results. Many laboratories that try miniaturizing assays use expensive automated liquid handlers to minimize human error. With the method proposed here, one microliter is the lowest volume that needs to be pipetted. This can be easily done with accurately calibrated pipettes.

The preparation of sequencing libraries is a costly and time-consuming component of next-generation sequencing. We describe a method for miniaturizing reaction volumes as well as reducing the time to generate SARS-CoV-2 sequences when using the Illumina DNA Prep

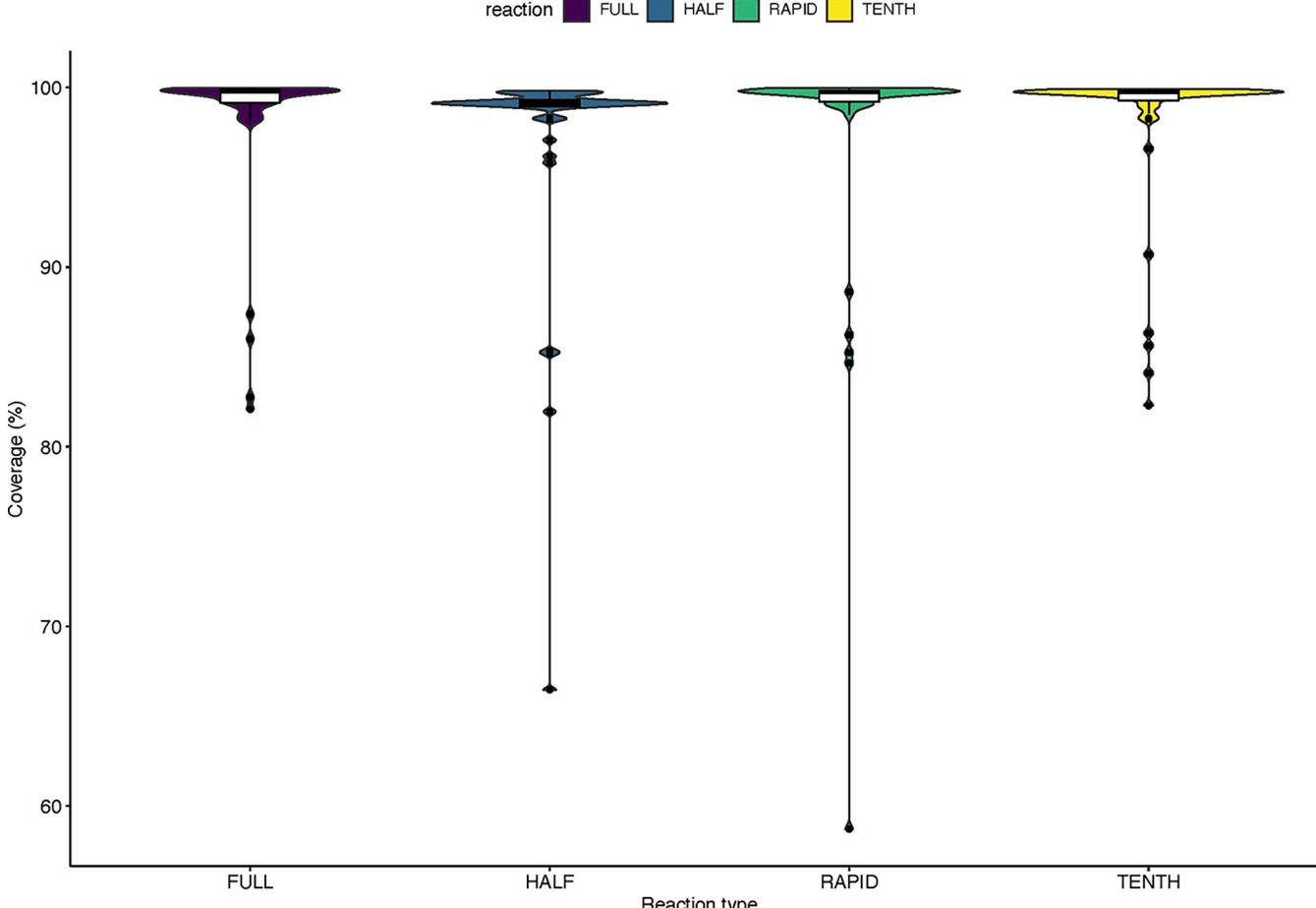

**Fig 2. Sequencing coverage of the four methods used.**

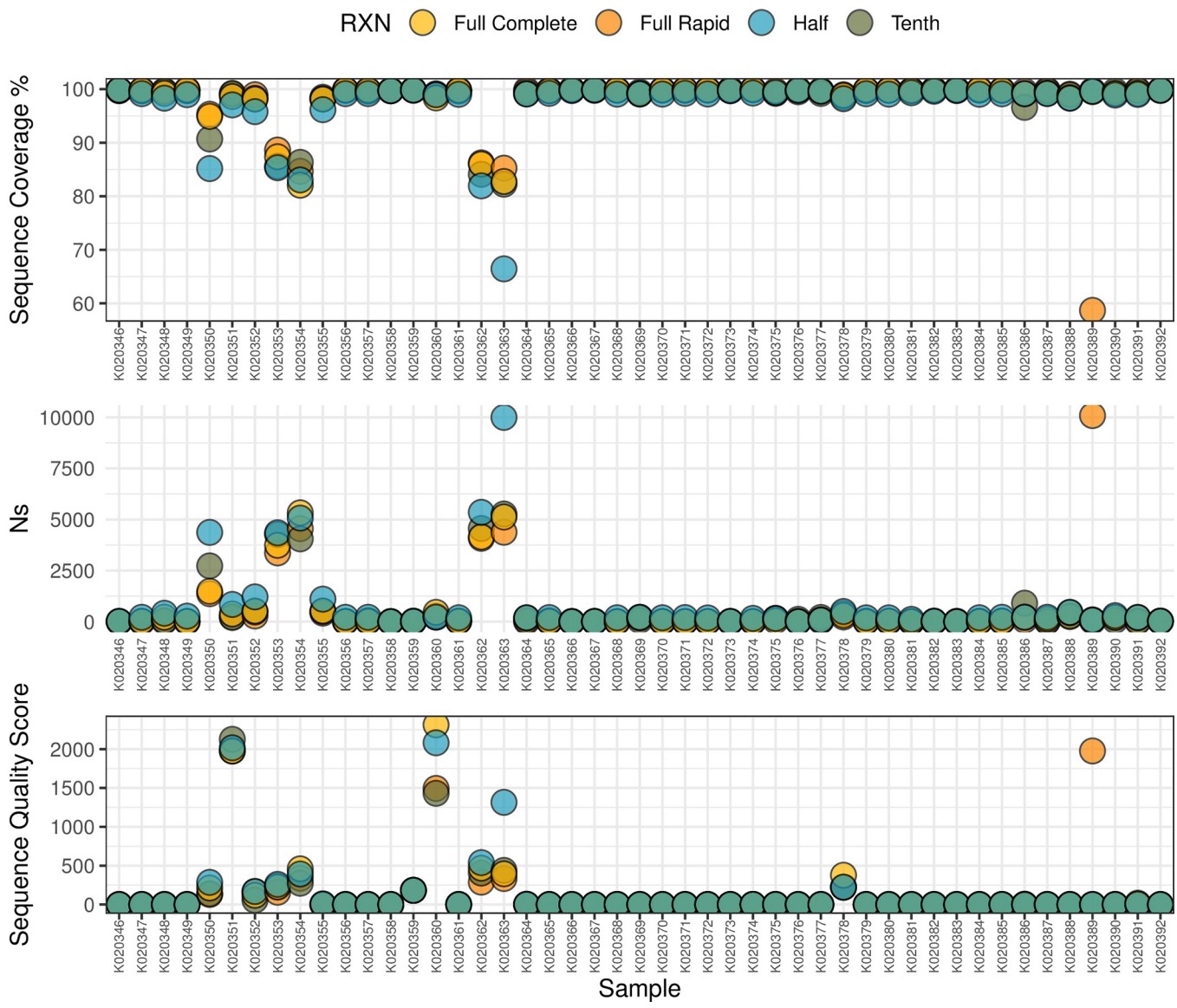

**Fig 3. Scatter plot showing the sequence coverage % and number of Ns for each sample in this experiment.**

library kit (Illumina). This detailed protocol has applicability in resource-constrained settings. There is no published evidence that this was done previously using this kit and in addition that it was manually done without an automated liquid handler. In this study, we show that a reduction in reaction volumes to as low as one-tenth volume does not compromise sequence coverage and quality. In recent years, the overall cost has significantly dropped but the cost of the reagents remains high [12].

In Africa, the cost of sequencing is relatively high due to the lower demand for sequencing compared to the rest of the world. Therefore, the ability to use volumes as low as one-tenth of the reaction volumes will afford an appreciable reduction in reagents and costs, therefore allowing for more samples to be processed with the same quantity of supplied reagents [11]. This can help reduce costs for the upstream processes of sequencing. In addition to the reduction of the library preparation reagents, we have also shown that using LunaScript for cDNA

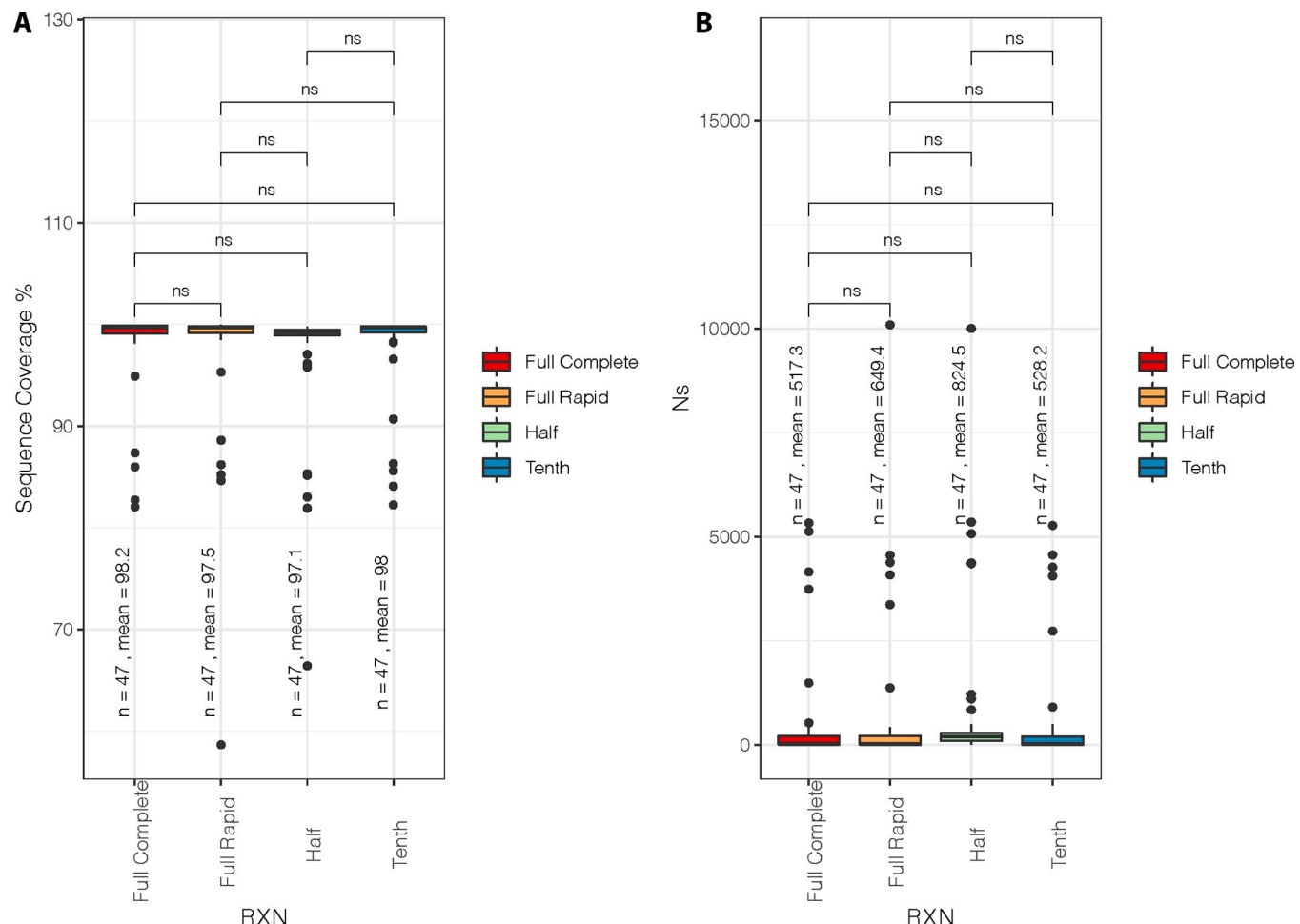

**Fig 4. Boxplots showing the sequence coverage %, and number of Ns for the different experimental categories, with the number of data points, and means of each category labelled.** The statistical significance results are also shown for each pairwise comparison (ns: non-significant). The boxes indicate the median (middle line) and the interquartile ranges (box edges) for each category. RXN describes the reaction, four different reactions were tested.

synthesis at a reduced volume and halving the reaction of the tiling PCR reagents, increases the cost-effectiveness overall. This miniaturized protocol (one-tenth reaction) reduced the cost for the Illumina DNA Prep library preparation by USD66 per sample; a drastic reduction in the cost. In addition, the time from extraction to sequencing for the one-tenth rapid reaction is reduced by 3.5 hours. To further reduce costs, ancillary equipment such as multichannel pipettes, rubber mats, and plate magnets are needed.

In this study, we used Nextclade to determine the quality profiles of the genomes produced. Nextclade is part of the Nextstrain host of tools that have been pivotal in the real-time tracking of SARS-CoV-2 and other infectious disease pathogens globally. In the case of SARS-CoV-2, Nextstrain predefined criteria for the quality of genomes to be included in downstream analysis. We used the same criteria as cut-offs for our sequencing success [28]. Briefly, the criteria place limits on the amount of missing data (Ns), number of private mutations, stop codons, frameshifts, mutation clusters and mixed sites (i.e., containing IUPAC characters). Based on these criteria, all our sequences from the one-tenth rapid method successfully passed the QC. We are therefore confident that our method not only mimics the full method but also yields genomes that are suitable for genomic surveillance analyses.

We also observed a high level of agreement across the different methods especially the full and the tenth reactions. There was only a single difference in the lineages assigned even though the coverages were comparable. This particular sample together with the other samples was treated alike. However, the Ct score was relatively high. We considered the one-tenth reaction as it produced sequences that were closest to those produced by the full reaction and enabled us to save the most amount of reagents—which directly translates into cost reduction. The time to process all of the rapid reactions was the same and therefore, did not impact the choice of the reaction except in the contrast with the full reaction.

## Limitations

This validation study was only based on a small number of samples and therefore did not allow us to assess the performance for the full and half reaction rapid against multiple sequencing runs. However, the tenth reaction that is the focus of this manuscript has since been used successfully in several laboratories in South Africa and shared with collaborating laboratories that have been conducting SARS-CoV-2 sequencing during the COVID-19 pandemic. The method has consistently yielded a success rate (high-quality genomes with coverage greater than 80%) of at least 70% in this study. This study relied on manual pipetting, which is subject to human variability and wastage. While this offers an advantage to small and medium-sized laboratories that cannot afford the high costs of procuring and maintaining liquid handling robots, the method could be easily set up on a liquid handling system. We have recently acquired liquid handling robots and are in the process of evaluating the variation in accuracy compared to the manual set-up. This method was only evaluated in one lab with one experienced medical scientist. We would like to carry out this same evaluation in different laboratories, by medical scientists with different levels of experience to ensure that comparable results can be achieved under different conditions.

Ideally, the same instrument should have been used to run the libraries for all four protocols, thus reducing the differential performance of the instrument affecting the results.

## Conclusion

This study has demonstrated that the reduction of reagent volumes using the Illumina DNA Prep kit has no major impact on the outcomes of the sequencing. The modified protocol could be readily adapted in countries that find it too expensive to sequence and routinely send their samples abroad for sequencing. This method can also be applied to other sequencing applications using the Illumina DNA Prep library kit eg. HIV and TB sequencing. The impact of the cost reduction will greatly benefit resource-limited countries where it is challenging to procure reagents. The emergence of SARS-Cov-2 has been a harsh lesson to the world for emerging diseases, our group has been developing these miniaturized procedures to aid in the surveillance of such emerging diseases. This protocol can be adapted for other pathogens such as Ebola, Zika, and Chikungunya for whole genome sequencing. This could greatly enhance the turn-around times for both genomic surveillance as well as treatment strategies.

## Supporting information

**S1 Table.**
(XLSX)

**S2 Table. All artic SARS-CoV-2 primers used in this study.**
(DOCX)

**S1 Fig.**
(PDF)

## Acknowledgments

The authors would like to acknowledge the Next Generation Sequencing Network of South Africa (NGS-SA) which provided the WGS information used in this study.

## Author Contributions

**Conceptualization:** Sureshnee Pillay, Tulio de Oliveira, Jennifer Giandhari.

**Data curation:** Sureshnee Pillay, Derek Tshiabuila, Richard J. Lessells, Jennifer Giandhari.

**Formal analysis:** Sureshnee Pillay, James Emmanuel San, Jennifer Giandhari.

**Funding acquisition:** Tulio de Oliveira.

**Investigation:** Sureshnee Pillay, James Emmanuel San, Derek Tshiabuila, Yeshnee Naidoo, Yusasha Pillay, Ugochukwu J. Anyaneji, Houriiyah Tegally, Jennifer Giandhari.

**Methodology:** Sureshnee Pillay, Jennifer Giandhari.

**Supervision:** Tulio de Oliveira, Jennifer Giandhari.

**Visualization:** Derek Tshiabuila, Houriiyah Tegally.

**Writing – original draft:** Sureshnee Pillay, Jennifer Giandhari.

**Writing – review & editing:** Sureshnee Pillay, James Emmanuel San, Derek Tshiabuila, Yeshnee Naidoo, Yusasha Pillay, Akhil Maharaj, Ugochukwu J. Anyaneji, Eduan Wilkinson, Houriiyah Tegally, Richard J. Lessells, Cheryl Baxter, Tulio de Oliveira, Jennifer Giandhari.

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
