## [Decision Letter · Decision Letter 0]

1 Dec 2022

PONE-D-22-30328Evaluation of miniaturized Illumina DNA preparation protocols for SARS-CoV-2 whole genome sequencingPLOS ONE

Dear Dr. Pillay,

Thank you for submitting your manuscript to PLOS ONE. After careful consideration, we feel that it has merit but does not fully meet PLOS ONE’s publication criteria as it currently stands. Therefore, we invite you to submit a revised version of the manuscript that addresses the points raised during the review process.

The authors are required to prepare a substantiated response to the reviewers' comments and a revised text of the manuscript in accordance with these comments.

We look forward to receiving your revised manuscript.

Kind regards,

Ruslan Kalendar

Academic Editor

PLOS ONE

“TdO- Rockefeller Foundation (HTH 017),  Abbott Pandemic Defense Coalition (APDC), the African Society for Laboratory Medicine, the National Institute of Health USA (U01 AI151698) for the United World Antivirus Research Network (UWARN) and the INFORM Africa project through IHVN (U54 TW012041), H3BioNet Africa (Grant # 2020 HTH 062), the South African Department of Science and Innovation (SA DSI) and the South African Medical Research Council (SAMRC) under the BRICS JAF #2020/049 and the World Bank (TF0B8412).

JG- South African Medical Research Council (MRC SIR-HIVDR POC)”

Please respond by return e-mail so that we can amend your financial disclosure and competing interests on your behalf.

Reviewers' comments:

Reviewer's Responses to Questions

**Comments to the Author**

1. Is the manuscript technically sound, and do the data support the conclusions?

Reviewer #1: Partly

Reviewer #2: Yes

Reviewer #3: Yes

Reviewer #4: Yes

2. Has the statistical analysis been performed appropriately and rigorously? 

Reviewer #1: No

Reviewer #2: Yes

Reviewer #3: Yes

Reviewer #4: Yes

3. Have the authors made all data underlying the findings in their manuscript fully available?

Reviewer #1: No

Reviewer #2: Yes

Reviewer #3: No

Reviewer #4: No

4. Is the manuscript presented in an intelligible fashion and written in standard English?

Reviewer #1: Yes

Reviewer #2: Yes

Reviewer #3: Yes

Reviewer #4: Yes

5. Review Comments to the Author

Reviewer #1: 

This manuscript by Sureshnee Pillay et al. describes a new (umpteenth) miniaturization method for preparing libraries for sequencing.

This manuscript is severely lacking in novelty to be published. In addition, the statistical comparisons are missing, for the most part, and the manuscript can be corrected according to my recommendations below.

Line 104/284: "to date"/"at the time" should be replaced by the date.

"et al." should be italicized.

Global: the manuscript should be turned to the passive.

Method: how was the number of samples to be sequenced determined (other than 48*2=96 for the indexes I guess?). Explanation of how the libraries are matched for sequencing is missing.

Present (at least in appendix) the primers used.

Websites must be referenced in the bibliography and not in the body of the text.

The experience of the operator has nothing to do in the body of the methods (possibly in the discussions, and even then it is not very useful ...)

How do the authors explain the difference in quality of a quarter protocol, when a tenth protocol worked better? Also, there is a lack of statistical comparisons to get an idea.

The sequences produced must be published (NCBI?) to be able to validate the results obtained.

Reviewer #2: 

This article is a research article that describes SARS-CoV-2 miniaturized sequencing protocol. CODVI-19 related research is of importance, especially an effective and cheap way of DNA sequence for future potential variants. I recommended acceptance after major revision.

1. Abstract should be a single paragraph instead of including four bullet points, since Introduction, Methods, Results, and Conclusion will be discussed in the main text. My suggestion is to polish and merge them to one paragraph less than 300 words.

2. Lack of comprehensive literature review on current DNA sequencing technology using other methods.

3. Figure quality is low. My suggestion is to increase the resolution and enhance the pictures. Also, bad color selections in figure 3, since Orange and Beige colors are not easy to tell the difference to many audiences, including me. My suggestion is to use a high-contrast color combination in figure design, for example the color choice in figure 4. Actually, keeping the color selection consistently is a good strategy to help audience understand the whole article by receiving the same color pattern information.

4. Some spelling and grammar mistakes. My suggestion is to perform more proof-reading.

Reviewer #3: 

The authors describe three modified sequencing protocols using Illumina platforms with the aim to reduce time, reagents and cost for SARS-CoV-2 surveillance. Some points need to be addressed.

1. Sequence data (FastQ) obtained by the three protocols need to be free accessible to the scientific community. I strongly suggest depositing these data on free accessible databases (es. Sequence Read Archive).

2. A phylogenetic analysis based on a Maximum likelihood approach that highlights the reproducibility of the three protocols by comparing the sequence data needs to be addressed.

3. It is important to provide evidence of reproducibility of the protocols across lineages (Omicron sublineages as latest). At this regard, the authors did not mention the lineage of the 47 samples used.

4. The authors should discuss the advantages that these methods can have in other settings, like other emerging and re-emerging pathogens surveillance (es. Ebola).

Reviewer #4: 

The study evaluation of different volume of library for SARS-CoV-2. Detailed library preparation kit is not provided, Nextera XT, or DNA flex ??? The detailed procedure/SOP needs to be deposited on to the Protocols.IO platform and link it to the manuscript. The NGS raw fastq files needs to be deposited to PubMed.

Lines 179-206, which kit used in the study, please provide the detailed information on it.

Table 2, how much of amplicon needed in stead of volume need also to be provided.

Lines 208-233, please provide a table to summarize the difference of non-rapid and rapid method process.

6. PLOS authors have the option to publish the peer review history of their article (what does this mean?). If published, this will include your full peer review and any attached files.

Reviewer #1: No

Reviewer #2: No

Reviewer #3: No

Reviewer #4: No

---

## [Author Response · Author response to Decision Letter 0]

31 Jan 2023

The study did not involve human subjects, therefore informed consent was not required. As mentioned in the manuscript lines lines 211-218, remnant samples were used from the National Health Laboratory Service as part of a SARS-CoV-2 genomic surveillance programme. Ethical approval for genomic surveillance performed in this study was obtained from the university of Kwazulu Natal IRB approval number BREC/00001510/2020.

“TdO- Rockefeller Foundation (HTH 017), Abbott Pandemic Defense Coalition (APDC), the African Society for Laboratory Medicine, the National Institute of Health USA (U01 AI151698) for the United World Antivirus Research Network (UWARN) and the INFORM Africa project through IHVN (U54 TW012041), H3BioNet Africa (Grant # 2020 HTH 062), the South African Department of Science and Innovation (SA DSI) and the South African Medical Research Council (SAMRC) under the BRICS JAF #2020/049 and the World Bank (TF0B8412).

JG- South African Medical Research Council (MRC SIR-HIVDR POC)”

Please respond by return e-mail so that we can amend your financial disclosure and competing interests on your behalf.

All the raw sequence data generated for this study has been made publicly available on the NCBI short read archive (SRA) under project no. PRJNA926488.

5. Review Comments to the Author

Reviewer #1: 

This manuscript by Sureshnee Pillay et al. describes a new (umpteenth) miniaturization method for preparing libraries for sequencing.

This manuscript is severely lacking in novelty to be published. In addition, the statistical comparisons are missing, for the most part, and the manuscript can be corrected according to my recommendations below.

Line 104/284: "to date"/"at the time" should be replaced by the date.

"et al." should be italicized.

We have updated the manuscript according to the reviewers recommendation. We added the appropriate timing. See Page 4, line 87 and page 14 line 487-488..

Global: the manuscript should be turned to the passive.

As recommended by the reviewer, we have changed the active sections of the manuscript to passive, for instance;

page 5, lines 166-167, have changed from “As a result, protocols have been modified to maximize reagent use” to “ As a result of limited supplies, protocol modification has been adopted as means to maximize reagent use (10) and drive down sequencing costs”.

page 6, lines 187-188, have changed from “”we sought to adapt our library preparation protocol into a lower-cost, streamlined protocol to “ in this study, an adaption of our library preparation protocol into a lower cost, streamlined protocol is described”.

Method: how was the number of samples to be sequenced determined (other than 48*2=96 for the indexes I guess?). Explanation of how the libraries are matched for sequencing is missing.

Yes, the reviewer is right, to manage the cost of sequencing, we split for methods across two sequencing runs. Each run accounted for 2 methods each having 48 samples. Importantly, the method of interest in this manuscript has since been used across several sequencing runs successfully which gives us further confidence in its performance and accuracy.

Present (at least in appendix) the primers used.

The table of primers has been added as supplementary table 1.

Websites must be referenced in the bibliography and not in the body of the text.

Please see below:

P13-Line 447

P24- line 704

The experience of the operator has nothing to do in the body of the methods (possibly in the discussions, and even then it is not very useful ...)

We agree with the reviewer and have removed this line from the text.

How do the authors explain the difference in quality of a quarter protocol, when a tenth protocol worked better? Also, there is a lack of statistical comparisons to get an idea.

In this study, we tested the full reaction as per manufactures instructions against the rapid protocol for the full, half and tenth reactions. We did not test the quarter. Although all the reactions performed well, in this manuscript we focused on the 10th reaction as it offered the most saving on the laboratory reagents and produced the sequences that were closest to the full reaction. We however, find this a interesting observation that would require validating across additional runs as we only tested the half reaction in one sequencing run. We have highlighted this in the limitations section of our manuscript and could be a basis for a larger validation study.

The sequences produced must be published (NCBI?) to be able to validate the results obtained. 

As requested, all the raw sequence data generated for this study has been made publicly available on the NCBI short read archive (SRA) under project no. PRJNA926488.

Reviewer #2: 

This article is a research article that describes SARS-CoV-2 miniaturized sequencing protocol. CODVI-19 related research is of importance, especially an effective and cheap way of DNA sequence for future potential variants. I recommended acceptance after major revision.

1. Abstract should be a single paragraph instead of including four bullet points, since Introduction, Methods, Results, and Conclusion will be discussed in the main text. My suggestion is to polish and merge them to one paragraph less than 300 words.

The abstract has been merged as per the reviewers recommendation: Page 1, lines 43-70. 

2. Lack of comprehensive literature review on current DNA sequencing technology using other methods.

Content added: page 4-5: line 90-162.

3. Figure quality is low. My suggestion is to increase the resolution and enhance the pictures. Also, bad color selections in figure 3, since Orange and Beige colors are not easy to tell the difference to many audiences, including me. My suggestion is to use a high-contrast color combination in figure design, for example the color choice in figure 4. Actually, keeping the color selection consistently is a good strategy to help audience understand the whole article by receiving the same color pattern information.

Figure 3. has been re-done with different colour selections, thus enhancing the quality, as per the reviewers recommendations.

4. Some spelling and grammar mistakes. My suggestion is to perform more proof-reading.

This has been addressed.

Reviewer #3: 

The authors describe three modified sequencing protocols using Illumina platforms with the aim to reduce time, reagents and cost for SARS-CoV-2 surveillance. Some points need to be addressed.

1. Sequence data (FastQ) obtained by the three protocols need to be free accessible to the scientific community. I strongly suggest depositing these data on free accessible databases (es. Sequence Read Archive).

As suggested by the reviewer, all the raw sequence data generated for this study has been made publicly available on the NCBI short read archive (SRA) under project no. PRJNA926488.

2. A phylogenetic analysis based on a Maximum likelihood approach that highlights the reproducibility of the three protocols by comparing the sequence data needs to be addressed.

A maximum likelihood phylogenetic tree has been inferred in IQTREE2 using the GTR evolutionary model and 100 boostrap replicates. The inferred tree is available as supplementary figure 1. As shown in the tree, we see a high level of congruence and similarity between the sequences from each method.

3. It is important to provide evidence of reproducibility of the protocols across lineages (Omicron sublineages as latest). At this regard, the authors did not mention the lineage of the 47 samples used.

This is information is provided in Table S1 and discussed on page 17: lines 547-550.

4. The authors should discuss the advantages that these methods can have in other settings, like other emerging and re-emerging pathogens surveillance (es. Ebola).

Page 20: lines 635-643.

Reviewer #4: 

The study evaluation of different volume of library for SARS-CoV-2. Detailed library preparation kit is not provided, Nextera XT, or DNA flex ??? 

Illumina DNA Prep- line 231

The detailed procedure/SOP needs to be deposited on to the Protocols.IO platform and link it to the manuscript. 

Line 231- dx.doi.org/10.17504/protocols.io.n92ldpp8nl5b/v1

The NGS raw fastq files needs to be deposited to PubMed.

As suggested by the reviewer, all the raw sequence data generated for this study has been made publicly available on the NCBI short read archive (SRA) under project no. PRJNA926488.

Lines 179-206, which kit used in the study, please provide the detailed information on it.

Nextera DNA Prep_line 231.

Table 2, how much of amplicon needed in stead of volume need also to be provided.

DNA was used neat and undiluted.

Lines 208-233, please provide a table to summarize the difference of non-rapid and rapid method process.

Information is provided in Table 3.

---

## [Decision Letter · Decision Letter 1]

6 Mar 2023

Evaluation of miniaturized Illumina DNA preparation protocols for SARS-CoV-2 whole genome sequencing

PONE-D-22-30328R1

Dear Dr. Pillay,

We’re pleased to inform you that your manuscript has been judged scientifically suitable for publication and will be formally accepted for publication once it meets all outstanding technical requirements.

Kind regards,

Ruslan Kalendar

Academic Editor

PLOS ONE

---

## [Editor Report · Acceptance letter]

18 Apr 2023

PONE-D-22-30328R1 

Evaluation of miniaturized Illumina DNA preparation protocols for SARS-CoV-2 whole genome sequencing 

Dear Dr. Pillay:

I'm pleased to inform you that your manuscript has been deemed suitable for publication in PLOS ONE. Congratulations! Your manuscript is now with our production department. 

Kind regards, 

on behalf of

Professor Ruslan Kalendar 

Academic Editor

PLOS ONE